# Recycled Untreated Rubber Waste for Controlling the Alkali–Silica Reaction in Concrete

**DOI:** 10.3390/ma15103584

**Published:** 2022-05-17

**Authors:** Safeer Abbas, Ali Ahmed, Ayesha Waheed, Wasim Abbass, Muhammad Yousaf, Sbahat Shaukat, Hisham Alabduljabbar, Youssef Ahmed Awad

**Affiliations:** 1Department of Civil Engineering, University of Engineering and Technology, Lahore 54890, Pakistan; safeer.abbas@uet.edu.pk (S.A.); ali@uet.edu.pk (A.A.); ayeshawaheed3@gmail.com (A.W.); yousaf_dr786@uet.edu.pk (M.Y.); 2Punjab Intermediate Cities Improvement Investment Program, Lahore 54660, Pakistan; sbahat.safeer@gmail.com; 3Department of Civil Engineering, College of Engineering in Al-Kharj, Prince Sattam Bin Abdulaziz University, Al-Kharj 11942, Saudi Arabia; h.alabduljabbar@psau.edu.sa; 4Structural Engineering Department, Faculty of Engineering & Technology, Future University in Egypt, New Cairo 11835, Egypt; youssef.ahmed@fue.edu.eg

**Keywords:** recycled rubber waste, alkali–silica reaction, ASR environment, expansion

## Abstract

Recycled rubber waste (RW) is produced at an alarming rate due to the deposition of 1.5 billion scrap tires annually around the globe, which causes serious threats to the environment due to its open land filling issues. This study investigates the potential application of RW in concrete structures for mitigating the alkali–silica reaction (ASR). Various proportions of RW (5%, 10%, 15%, 20%, and 25%) partially replaced the used aggregates. RW was procured from a local rubber recycling unit. Cubes, prisms, and mortar bar specimens were prepared using a mixture design recommended by ASTM C1260 and tested for evaluating the compressive and flexural strengths and expansion in an ASR conducive environment for specimens incorporating RW. It was observed that the compressive and flexural strength decreased for specimens incorporating RW compared to that of the control specimens without RW. For example, an 18% and an 8% decrease in compressive and flexural strengths, respectively, were observed for specimens with 5% of RW by aggregates volume at 28 days. Mortar bar specimens without RW showed an expansion of 0.23% and 0.28% at 14 and 28 days, respectively, indicating the potential ASR reactivity in accordance with ASTM C1260. A decrease in expansion was observed for mixtures incorporating RW. Specimens incorporating 20% of RW by aggregate volume showed expansions of 0.17% at 28 days, within the limit specified by ASTM C1260. Moreover, specimens incorporating RW showed a lower reduction in compressive and flexural strengths under an ASR conducive environment compared to that of the control specimen without RW. Micro-structural analysis also showed significant micro-cracking for specimens without RW due to ASR. However, no surface cracks were observed for specimens incorporating RW. It can be argued that the use of RW in the construction industry assists in reducing the landfill depositing issues with the additional benefit of limiting the ASR expansion.

## 1. Introduction

The Alkali–silica reaction (ASR) is the main deleterious process responsible for premature deterioration in concrete structures. It is a chemical reaction between alkalis present in the cement matrix and reactive siliceous phases in the aggregates. Due to this reaction, alkali–silica gel will form that builds pressure and creates stresses on the surrounding concrete under special circumstances, including the availability of moisture and the exposure to elevated temperatures [1]. Various visual symptoms of ASR affected structures include cracking, localized concrete crushing, relative expansion and deformation, surface staining, gel exudations, and surface pop-outs [2]. Various structures were reported in the past that were damaged due to ASR around the world (Table 1). Similarly, hydraulic structures (Warsek and Terbela dams) in Pakistan were damaged due to ASR [3].

**Table 1 materials-15-03584-t001:** Examples of ASR damaged structures around the world.

Sr. No.	Structures	Location	References
1	Retaining wall and piers	Japan	Ono [4]
2	turbine foundation, Ikata power plant	Japan	Takakura et al. [5]
3	Bridge piers of Hanshin Super highway	Japan	Miyagawa et al. [6]
4	Hokuriku Expressway	Japan	Moriyama and Nomura [7]
5	Bibb Graves Bridge	Alabama, USA	Michael et al. [8]
6	Concrete pavement	Arkansas and Delaware, USA	
7	Concrete abutments	Rhode Island and Maine, USA	
8	Concrete barrier walls	Massachusetts, USA	
9	Bridge columns	Houstan, Texas, USA	
10	Seabrook Nuclear Power Plant	New Hampshire, USA	Saouma et al. [9]
11	Flyover bridge	Aarhus, Denmark	Schmidt et al. [10]
12	Vosnasvej Bridge	Denmark	Gustenhoff et al. [11]
13	Robert-Bourassa/Charest overpass	Quebec City, Canada	Sanchez et al. [12]
14	Bridge structures	Finland	Lahdensivu et al. [13]

Different strategies have been investigated in the past for mitigating the ASR. For example, supplementary waste materials such as slag, silica fume, coal fly ash, metakaolin, rice husk ash, glass powder, and sugarcane bagasse ash have been incorporated in concrete mixtures for controlling the ASR expansion [14,15,16,17,18,19,20,21]. Actually, supplementary cementitious materials (SCMs) were used as a partial replacement of cement that resulted in a decrease in alkali contents in the mixture for ASR to initiate. Furthermore, SCMs help in reducing the overall porosity limiting the moisture penetration into the concrete mixture for accelerating the ASR [22,23]. Thomas [14] conducted field studies and concluded that the concrete mixtures with fly ash showed an improved performance against ASR-associated damages. Abbas et al. [19] reported a 23% and a 50% decrease in mortar bar expansion due to an incorporation of 10% and 40% of rice husk ash by cement weight. Approximately, a 20% and a 40% decrease in ASR expansion was reported for mixtures with 10% and 40% of sugarcane bagasse ash [20]. Similarly, ASR expansion was decreased when cement was partially replaced with waste marble powder [24]. Other studies used lithium admixtures for mitigating ASR expansion [25,26,27,28]. Furthermore, fiber reinforcement can also be used for improved properties of cementitious composites [29].

Recycled rubber waste (RW) is generated from scrap vehicular rubber tires. Around 1.5 billion tires are disposed of or buried each year, which may reach 5 billion by 2030 [30,31]. These waste rubber tires cause black pollution and serious panic to environment, due to its easy disposal through the burning process that causes fire and smoke hazards and other harmful emissions that disturb the natural eco-system and threaten human health [32]. Therefore, it is necessary to find an alternative way of disposing the waste tires for an improved ecological balance. RW can be used in concrete mixtures as a partial replacement of aggregate with the added value benefit for mitigating the ASR in concrete structures. RW in concrete may be used in the form of rubber chips, rubber crumb, and rubber powder [33]. Various previous studies have been conducted to study the mechanical and durability properties of concrete incorporating RW [34,35,36,37,38,39,40]. For instance, Liu et al. [41] reported a 20% reduction in compressive strength for specimens with 15% of crumb rubber. Similarly, a study conducted by Batayneh et al. [42] showed a 50% decrease in compressive strength due to the incorporation of 40% rubber. Bing and Ning [43] reported a 26% decrease in compressive strength for specimens incorporating 25% of rubber particles. It should be noted that the compressive strength of mixtures incorporating rubber waste was lower than the control mixture; however, they showed an improved energy absorption ability leading to ductile failure [44]. 

Limited previous studies are available on the use of untreated recycled rubber waste (RW) for mitigating the ASR-associated damages in concrete infrastructures. Therefore, this study explores the various proportions of RW partially replacing reactive aggregates for the possible mitigation of ASR expansion. Furthermore, strength properties of specimens incorporating various proportions of RW were investigated and compared with ASR conditioning. The novelty of this research is to demonstrate the use of RW for improved concrete performance with regard to its durability properties. Furthermore, the findings of this research will facilitate the stakeholders with the economical and sustainable use of RW in the concrete industry, leading to the avoidance of the open land filling with the additional benefit of limiting the ASR expansion and associated damages.

## 2. Materials and Casting of Specimens

Ordinary Portland cement was used. Recycled rubber waste (RW) and aggregates were procured from a local market. Aggregates were rinsed with water and dried in oven before crushing into various size fractions in accordance with ASTM C1260 [45]. Figure 1 shows the RW and used aggregates. Various proportions of RW were replaced with aggregates by weight (Table 2) and investigated for mechanical and durability properties. Normal water was used in the concrete mixture for mixing purposes. 

According to ASTM C1260 [45], a cement to aggregate ratio of 1 to 2.25 and a water to cement ratio of 0.47 was used for preparing various specimens. The size of cube, prism, and mortar bar specimens were 50 mm × 50 mm × 50 mm, 40 mm × 40 mm × 160 mm, and 25 mm × 25 mm × 285 mm, respectively. For each testing ages, five numbers of specimens were cast for tested mixtures. After 24 hours of casting, specimens were taken out from their respective molds and placed in water. After one day, specimens were paced in sodium hydroxide (NaOH) solution at 80 °C in accordance with ASTM C1260 [45]. Furthermore, other set of identical specimens were also cured in normal water at room temperature for comparison purposes with the ASR conditions. Figure 2 shows the specimen preparation process. 

## 3. Experimental Methodologies

Initially, various tests on used raw materials were performed for investigating their physical and chemical properties. For instance, X-ray fluorescence (XRF) and X-ray diffraction (XRD) analyses were performed on used cement, recycled rubber waste (RW), and aggregates. Micro-structural analysis of used cement and RW was conducted using Zeiss, Germany field emission scanning electron microscopy (SEM) and energy dispersive X-rays (EDX). Physical properties of aggregates were also determined by performing specific gravity, bulk density, water absorption, and impact resistance tests. Petrographic analysis on used aggregates was also conducted as per ASTM C295 [46]. Thermal analysis on untreated RW and powder concrete specimens incorporating various dosages of RW was performed by heating the specimens up to 1200 °C and measuring the heat flow and mass change. 

Fresh properties of mixtures incorporating cement, aggregates, and RW were measured by performing the flow test using ASTM C1437 [47]. Setting times and standard consistency were determined on tested mixtures according to ASTM C191 [48] and ASTM C187 [49], respectively.

Compressive and flexural strengths on cube and prismatic specimens were determined at 7, 14, 28, 56, 90, 120, and 150 days as per ASTM C109 [50] and ASTM C348 [51], respectively. Loading was applied on tested specimens at a rate of 990 N/s for compression and 2600 N/s for flexural testing. Density of cube specimens was also determined using the measured mass and volume values of respective specimens. Expansion measurements at 3, 7, 14, 21, 28, 42, 56, 90, 120, and 150 days were taken on mortar bar specimens using digital length comparator according to ASTM C490 [52] (Figure 2). Mortar bar specimens were also visually monitored for each time during measurements for any surface cracking and damages. SEM and EDX analyses were also conducted on various specimens for examining the internal micro-structures. 

## 4. Results and Discussion

### 4.1. Properties of Raw Materials

Table 3 shows the chemical composition of the used waste rubber. Carbon contents were the major component in the used rubber waste. A similar chemical composition of rubber waste was also mentioned in other studies [53,54]. Table 4 shows the chemical composition of the used cement. Figure 3 shows the SEM images of the used cement and RW. The particles of cement and RW were of an irregular shape and a non-uniform size. Moreover, the surface of RW particles were rougher (Figure 4) than the cement particles. The EDX analysis of RW shows the prominent peaks of carbon, calcium, and silicon (Figure 3d). The phase identification of cement performed through XRD analysis showed the presence of tri-calcium silicate (C_3_S), di-calcium silicate (C_2_S), and gypsum (Figure 5a). Similar XRD patterns of cement were also reported elsewhere [55]. Further, the carbon content peak was maximum in the XRD analysis of used RW (Figure 5b). The physical properties of used cement and RW are shown in Table 5. The specific gravity of RW was 1.24, which is lower than the used cement (3.15). The unit weights of cement and RW were 1427 and 285 kg/m^3^. The lower specific gravity and unit weight of RW will lead towards a lighter structure. Table 6 shows the physical and chemical properties of the used aggregates. The bulk density and specific gravity of the used aggregates were 1429 kg/m^3^ and 2.63, respectively. The impact value of aggregates was 2.81%. 

Chemical properties showed that the used aggregates had high silica contents. Figure 6 shows the petrographic analysis results of used aggregates. It was observed that the used aggregates composed of sandstone, siltstone, and shale (Figure 6a). Sandstone was brownish in color and the size was up to 0.60 mm. The sandstone was mainly consisted of quartz and feldspar (Figure 6b). The siltstone was grayish in color and consisted of quartz as discrete grains or a poly-crystalline nature. Feldspar also occurred in siltstone as K-feldspar. The mafic phases in siltstone incorporated the pyroxene, amphibole, and mica. The shale was reddish brown in color and consisted of clays (75%) and quartz (25%) (Figure 6d). Based on petrographic analysis, it can be concluded that the used aggregate was mainly composed of quartz (stained and polycrystalline) and shale components; therefore, it can be vulnerable to ASR. Hence, the used aggregate was suitable to access the potential of RW in repressing the ASR-associated damages. A detailed petrographic analysis can be found elsewhere [55].

### 4.2. Flow of Mixtures Incorporating Recycled Rubber Waste

Table 7 shows the flow results of the tested mixtures. The control mixture without RW showed a flow of 114 mm. It was observed that the mixture’s flow was decreased due to the incorporation of RW. For instance, the mixtures RW5 and RW15 exhibited flows of 110 mm and 105 mm, respectively. A decrease in flowability for the mixture incorporating RW may be attributed to increased friction between the rubber particles and the matrix due to the rougher surface of the rubber particles. 

Furthermore, a higher fineness of rubber waste in comparison with the used aggregates lead to a reduction in the flow properties. Abdelmonem et al. [56] reported an 8.5% decrease in slump for a mixture incorporating 30% of crumb rubber by volume of aggregates compared to that of the control mixture. Similarly, Taha et al. [57] reported around a 40% reduction in slump for a mixture with 50% of crumb rubber by volume of aggregates compared to identical mixture without rubber.

### 4.3. Density

Figure 7 shows the density results of the mixture incorporating RW. A decreasing trend in density was observed due to the incorporation of RW. For instance, around a 6% and 14% decrease in density at 28 days was observed for specimens with 10% and 20% of RW by volume of aggregates compared to that of the identical specimen without RW. This reduction in density due to the incorporation of RW was due to the lower specific gravity of used rubber particles in comparison with the aggregates. Further, mixtures incorporating RW have the ability to entrap air at the surface of the rubber particles due to its jagged texture leading to a reduction in the unit weight [57]. Bisht and Ramana [53] reported a 10% decrease in density due to the incorporation of 5.5% of crumb rubber by volume of aggregates. Abdelmonem et al. [56] reported around a 9% decrease in density for a specimen with 30% of crumb rubber by volume of aggregates. Similarly, Taha et al. [57] observed around a 10% reduction in density for a specimen with 25% of crumbed tire rubber compared to the control specimen without rubber.

### 4.4. Thermal Analysis

Figure 8 shows the thermal analysis of used untreated rubber waste (RW). Thermogravimetric analysis (TGA) of RW represented the weight loss over various temperatures and exhibited two distinct regions from 0 to 1200 °C (Figure 8). An approximate 22% weight loss of RW was observed from 0 to 1000 °C at a constant rate. This weight loss was related to the main ingredients of RW including oil, plasticizers, and other additives used for the manufacturing of rubber [57]. A sudden drop in weight was observed from 1000 °C to 1200 °C. A total weight loss of approximately 40% was noted from 0 to 1200 °C. Differential scanning calorimetry (DSC) of RW showed the presence of various peaks (Figure 8). Two endothermic peaks at around 138 °C and 1010 °C were observed in DSC analysis of RW, while exothermic peaks were noted at 325 °C and 1146 °C for rubber specimens.

Figure 9 shows, respectively, the TGA and DSC of specimens incorporating various dosages of RW. The TGA curve shows that the weight loss can be divided into three zones for specimens incorporating RW (Figure 9a): 0 to 400 °C, 401 to 700 °C, and 701 to 1200 °C. More mass loss was observed for specimens incorporating higher proportions of RW. 

For example, approximately 5.50%, 7.65%, and 15.02% losses in mass were observed for specimens incorporating 0%, 10%, and 25% of RW, respectively, in the temperature range of 0 to 400 °C. Similarly, mass losses of 4.50%, 7.95%, and 8.01% were observed for 0%, 10%, and 25% of RW specimens, respectively, in the temperature range of 401 to 700 °C. At 1200 °C, the total mass losses of specimens incorporating 0%, 5%, 10%, 15%, and 25% of RW were around 14%, 23%, 32%, 35%, and 47%, respectively. This mass loss was related to the initial loss of water in voids or micro-pores, the disintegration of hydrated compounds, and the burning of rubber and other organic matter. Figure 9b shows DSC curves for specimens incorporating various proportions of RW. It was noted that the general trend of heat flow versus temperature was comparable for the control specimen and specimens with RW (Figure 9b). It was observed that an initial endothermic peak was observed at around 86 °C for the control specimen without RW. It was mainly due to the evaporation of water in pores. Other endothermic peaks related to the decomposition of calcium hydroxide and calcium carbonates were observed at around 444 °C and 727 °C, respectively, for the control specimen. Similar observations were also reported in previous studies [19,20,24,58,59]. For specimens with 5% of RW, endothermic peaks were observed at around 87 °C, 466 °C, and 738 °C corresponding to a heat flow of 7, 10, and 27 mcal/sec, respectively. Similarly, specimens incorporating 15% of RW showed endothermic peaks at around 98 °C, 470 °C, and 741 °C (Figure 9b).

### 4.5. Effect of Recycled Rubber Waste on Compressive and Flexural Strengths

Figure 10a shows the effect of various proportions of RW on compressive strength results. All the compressive strength results reported were the average of five identical specimens having a coefficient of variance (COV) less than 3%. 

An increase in compressive strength with time was observed for all the tested specimens. For example, compressive strength for the control specimens were around 29 and 40 MPa at 28 and 120 days, respectively. A decrease in compressive strength was noted for specimens incorporating recycled rubber waste (RW) compared to that of the control specimens at all the tested ages. For instance, an approximate 18%, 24%, and 29% decrease in compressive strengths at 28 days for specimens incorporating 5%, 10%, and 15% of RW by volume, respectively, compared to the control specimens. Similarly, around a 22%, 31%, and 45% decrease in compressive strength for specimens incorporating 5%, 10%, and 20% of RW by volume of aggregate, respectively, at 90 days. A maximum decrease in compressive strength of around 47% was observed at 150 days for specimens incorporating 25% of RW compared to specimens without RW. Figure 11 shows the SEM image of a specimen incorporating 25% of RW, indicating the random distribution of rubber particles. The decrease in compressive strength due to the incorporation of RW was due to poor/weak bonds between the rubber particles and the matrix, confirmed through SEM analysis (Figure 12). It was reported that the cracks were initiated at the interfaces between the matrix and the rubber particles as the loading was applied on the specimens. As RW was partially replaced with aggregates, a decrease in compressive strength for mixtures with RW was therefore related to lower stiffness, softness, and the elastically deformable material of rubber particles, leading to a decrease in the overall load carrying capacity of concrete [53]. Furthermore, compressive strength loss for mixtures incorporating RW was also related to a decrease in density for specimens with rubber particles [57]. A similar decrease in compressive strength due to the incorporation of rubber was also reported in previous studies. For instance, Taha et al. [57] reported 15%, 25%, 50%, and 67% decreases in 28 days compressive strength for specimens with 25%, 50%, 75%, and 100% of crumb rubber by volume of aggregates, respectively, compared to control specimens. Similarly, Abdelmonem et al. [56] observed around 30%, 43%, and 47% decreases in 28-days compressive strength for specimens incorporating 10%, 20%, and 30% of rubber by volume of aggregates, respectively, compared to specimens without crumb rubber. Further decreases in compressive strength were observed for mixtures incorporating higher proportions of RW (Figure 10a). This can also be related to the development of more pores due to the finer nature of used RW. The finer material acts as a filler up to certain level of its utilization. However, beyond that certain limit, its addition will result in the development of voids. Similar observations were also mentioned in other studies [38,53,60,61]. 

Figure 10b shows the flexural strength of specimens incorporating RW. Results of flexural specimens showed a COV less than 2%. The control specimen without RW had a flexural strength of around 10 MPa at 28 days. A continuous decrease in flexural strength was observed for specimens incorporating RW. For instance, a decrease in the flexural strength of around 8%, 22%, and 35% was observed for specimens incorporating 5%, 10%, and 15% of RW by volume of aggregates compared to specimens with RW. A maximum decrease in flexural strength was reported for the mixture with 25% of RW. A decrease in flexural strength for specimens incorporating RW was dependent on the rubber particle shape and size [38]. The irregular nature of used RW will make the weaker bond between the rubber particles and the matrix. It should also be noted that the tested specimens were casted on a vibratory table that allows the water to stick to the rubber particles leading to a weakening of the interface. Furthermore, it was also reported in a previous study that the reduction in interfacial properties between rubber particles and the matrix can be attributed to the smoother texture of rubber particles [53]. This will result in the development of early cracks upon loading. Similar findings were also observed in previous studies [37,38,56]. Abdelmonem et al. [56] reported an approximate 3% and 28% decrease in flexural strength for specimens incorporating 10% and 30% of crumb rubber by volume of aggregates, respectively, compared to identical specimens without RW. Moreover, XRD patterns of mixtures incorporating various proportions of RW (Figure 13) indicated that no new phases in the hydration products were formed due to the inclusion of rubber particles. A similar finding was also observed in a previous study [33]. The major disadvantage of concrete mixtures incorporating recycled rubber waste (RW) is the poor matrix–rubber bond. Surface treatment of rubber particles will improve the rubber–matrix interface and hence, enhance the compressive and flexural strengths. 

### 4.6. Expansion Due to ASR

Expansion results of mortar bar specimens incorporating various proportions of RW are shown in Figure 14. Results shown in Figure 14 show the average value of five specimens with a COV less than 2%. The control specimen without RW showed an expansion of 0.232% and 0.284% at 14 days and 28 days. According to ASTM C1260, mortar bar specimens exposed in NaOH solution showing an expansion higher than 0.10% at 14 days and 0.20% at 28 days can be considered as alkali–silica reactive [45]. Similarly, according to RILEM AAR-2 [62], specimens can be categorized as reactive if the expansion at 16 days is greater than 0.10%. According to the Australian Standard AS-1141.60.1 [63], if specimens show an expansion less than 0.10% and 0.30% at 10 and 21 days, respectively, it can be assumed as low reactive. Further, if expansion is less than 0.10% at 28 days, specimens can be considered as non-reactive [63]. Therefore, it can be argued that the used aggregates were reactive in nature and exhibited alkali–silica reactivity. For specimens incorporating RW by volume of aggregates, a decrease in ASR expansion was observed compared to that of the identical specimen without RW. For example, specimens incorporating 5% of RW showed an expansion of 0.20% and 0.26% at 14 and 28 days, respectively, higher than the limit specified in ASTM C1260 and considered as reactive [45]. 

A further decrease in expansion was noted for specimens with higher proportions of RW. Specimens with 10% of RW showed an expansion of 0.175% at 14 days and 0.220% at 28 days. The expansion value was 0.20% at 28 days for specimens incorporating 15% of RW. Mixtures with 20% and 25% of RW by aggregate volume showed expansion values lower than 0.20% at 28 days, satisfying the ASTM C1260 limit for non-reactive ASR nature [45]. This reduction in ASR expansion due to the incorporation of RW was mainly due to the elastic behavior of rubber that ensures the dissipation of expansion stresses, initiated due to the formation of ASR gel. Similar observations were also stated in a previous study [44]. 

It was also observed that the expansion increased at later ages for all the tested specimens. For instance, specimens without RW exhibited expansions of 0.370% at 120 days and 0.284% at 28 days. Similarly, specimens incorporating 15% of RW showed expansion of 0.278% at 90 days and 0.195% at 21 days. Zhu [64] also reported that concrete mixtures incorporating RW as a partial replacement of aggregates will improve the concrete performance by introducing flexible particles into the concrete mixtures. Therefore, it can be argued that RW can be utilized in concrete for its efficient disposal relieving for the environmental overburden with the additional benefit of mitigating the ASR.

After performance tests, specimens were visually examined for surface cracking and other associated damages. Specimens without RW showed significant surface cracking, indicating the formation of ASR gel and the associated expansion (Figure 15a). Similar surface crack images for specimens exposed in ASR solutions were also reported elsewhere [65]. No significant surface cracks were observed in specimens incorporating RW (Figure 15b). Actually, rubber particles provided the crack arrestment mechanism that absorbed the associated increased stresses and limited the further penetration and formation of new cracks due to the ASR mechanism [44]. 

### 4.7. Effect of ASR on Compressive and Flexural Strengths

Figure 16a shows the reduction in compressive and flexural strengths for specimens incorporating various proportions of RW exposed to ASR conditions (80 °C in NaOH solution). A reduction in compressive strength was observed due to ASR exposure. For example, control specimens without RW subjected to ASR conditions showed decreases in compressive strength of 8%, 11%, and 15% at 28, 90, and 150 days, respectively, compared to that of the identical specimen under a normal curing regime. This reduction in compressive strength due to the ASR environment was mainly attributed to development of ASR gel, leading to formation of micro-cracks. A similar decrease in compressive strength was also described in previous studies [66,67]. 

It was also observed that a relatively higher reduction in compressive strength was noted for specimens incorporating RW. Maximum decreases in compressive strength of around 20% were observed at 150 days for specimens incorporating 25% of RW by volume of aggregates exposed to ASR conditions, compared to that of the identical specimen placed in normal curing. Wang [68] also reported a 20% decrease in compressive strength after one year of exposure to an ASR environment. Likewise, flexural strength was also decreased for specimens exposed to an ASR environment compared to that of the identical specimen placed in normal water (Figure 16b). For instance, control specimens without RW subjected to an ASR solution at 80 °C showed a 6% and 13% decrease in flexural strength at 56 and 150 days, respectively, compared to that of the identical specimens in water curing. 

### 4.8. Micro-Structural Analysis

Figure 12 shows the SEM images of specimens incorporating RW after 28 days under normal water curing. A weaker bond between rubber particles and the matrix was observed. Furthermore, micro-cracks were detected at the interface and around the rubber particles surroundings. These micro-cracks were developed due to increased tensile strain and the rubber particles being the softer material [57]. Furthermore, cracks in the rubber particles itself were also observed (Figure 12). Cracks in the rubber particles supported the fact that the stresses were transferred from the matrix to the rubber particles and subjected to tensile strain before failure. Similar observations were also reported in a previous study [57]. Mortar bar specimens exposed to an ASR-conducive environment were also analyzed using SEMs. The formation of ASR gel was observed in between the cement paste and aggregates for all the tested specimens. Furthermore, some micro-pores in the concrete mixture were also filled with ASR gel (Figure 17). Control specimens without RW showed a significant formation of micro-cracks, indicating the ASR distress (Figure 18). However, specimens with RW showed improved cracking behavior and micro-structure due to ASR (Figure 19). 

## 5. Conclusions

This study examined the behavior of untreated recycled rubber waste (RW) for mitigating ASR-associated damages. Cubes, prisms, and mortar bars incorporating various proportions of RW (0% to 25% by aggregate volume) were examined for compression, flexural, and expansion properties. Petrographic and micro-structural analyses were conducted to investigate the internal behavior of mixtures incorporating RW. Based on the conducted experiments, the following are the conclusions drawn: Mixtures incorporating RW showed decreased flow compared to that of the identical mixture without RW. For example, the control mixture exhibited a flow of 114 mm while the mixture with 15% of RW showed a flow of 105 mm. This was due to interfacial properties between the rubber particles and the matrix. Furthermore, the rougher surface of rubber particles hinders the flow properties of the tested mixtures.The density of mixtures with RW had shown a decreasing trend due to the lower value of the specific gravity of rubber particles. Furthermore, mixtures incorporating RW can entrap air at the surface of the rubber particles due to its jagged texture, leading to a reduction in the unit weight and density.Compressive strength was decreased for specimens incorporating RW compared to that of the identical specimens without RW at all the tested ages. For example, around a 24% decrease in compressive strength at 28 days was observed for specimens with 10% of RW by volume compared to that of the control specimen. This decrease was higher for specimens incorporating higher proportions of RW. A maximum compressive strength decrease of around 47% was observed at 150 days for specimens incorporating 25% of RW compared to identical specimens without RW. This reduction in compressive strength for specimens incorporating RW was attributed to weaker bonds between the rubber particles and the matrix.A decreasing trend in flexural strength was observed due to incorporation of RW. For instance, a decrease in flexural strength of around 8% was observed for specimens incorporating 5% of RW by volume of aggregates compared to that of the control specimens. A maximum decrease in flexural strength was reported for the mixture with 25% of RW. The irregular nature of used RW will make a weaker link between the rubber particles and the matrix, leading to a decrease in the flexural strength for specimens incorporating RW.Control specimens without RW showed expansions of 0.232% and 0.284% at 14 days and 28 days, respectively, indicating the reactive nature with respect to ASTM C1260. Expansion was decreased for specimens incorporating RW. For instance, mortar bars incorporating 5% of RW showed expansions of 0.20% and 0.26% at 14 days and 28 days, respectively. The mixture with 20% and 25% of RW by aggregate volume showed the expansion values lower than 0.20% at 28 days, satisfying the ASTM C1260 limit for non-reactive ASR nature. This decrease in ASR expansion for mixtures incorporating RW was mainly due to the elastic behavior of rubber that ensures the dissipation of expansion stresses, initiated due to the formation of ASR gel.The compressive and flexural strengths of specimens exposed to an ASR environment have shown a decreased behavior compared to identical specimens placed in water curing. For example, control specimens without RW subjected to ASR conditions showed a reduction in the compressive strength of 15% at 150 days, respectively, compared to that of the identical specimen under a normal curing regime. A maximum decrease in compressive strength of around 20% at 150 days was observed for specimens incorporating 25% of RW by volume of aggregates exposed to ASR conditions compared to that of the identical specimen placed in normal curing.Micro-structural analysis of specimens incorporating RW showed no surface cracking, although ASR gel formation was observed. However, control specimens without RW showed severe micro-cracks due to ASR.

It can be concluded that the use of RW as a partial replacement of reactive aggregates in the construction industry will reduce the landfilling issues of RW, while controlling the disastrous damages due to ASR in concrete infrastructures. It should also be noted that the pretreatment of RW can further improve the concrete properties leading to sustainable construction, which warrants further detailed future study. Furthermore, a future study using concrete prism expansion tests (ASTM C1293) needs to be conducted to compare and validate the experimental findings of this current research. 

## Figures and Tables

**Figure 1 materials-15-03584-f001:**
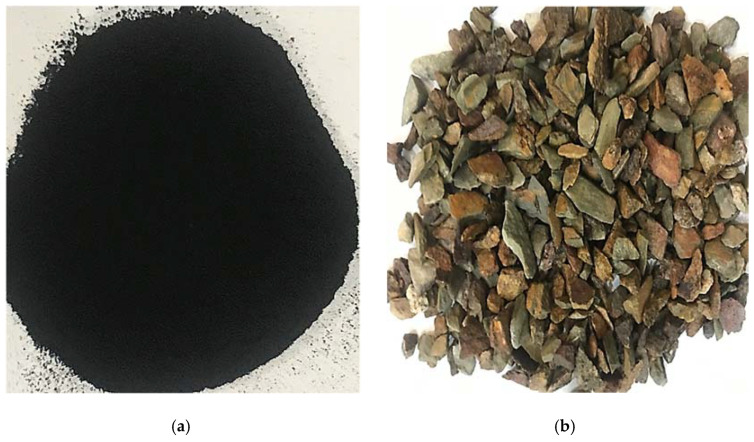
Materials used. (**a**) Recycled rubber waste; (**b**) Used aggregates.

**Figure 2 materials-15-03584-f002:**
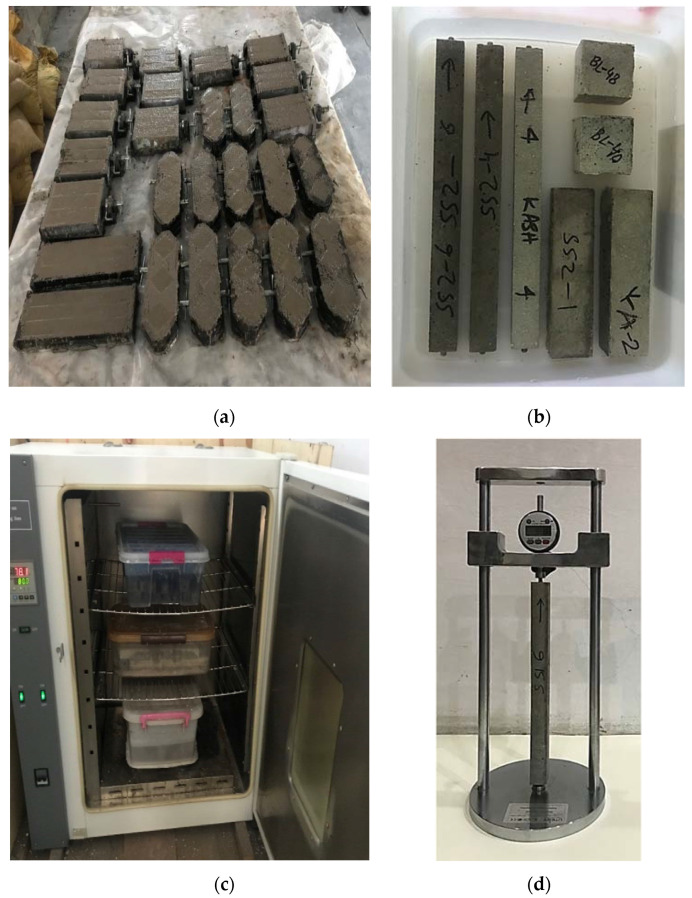
Preparation and testing of specimens. (**a**) Preparation of various specimens; (**b**) Specimens placed in solution; (**c**) Specimens placed in oven at 80 °C; (**d**) Digital length comparator.

**Figure 3 materials-15-03584-f003:**
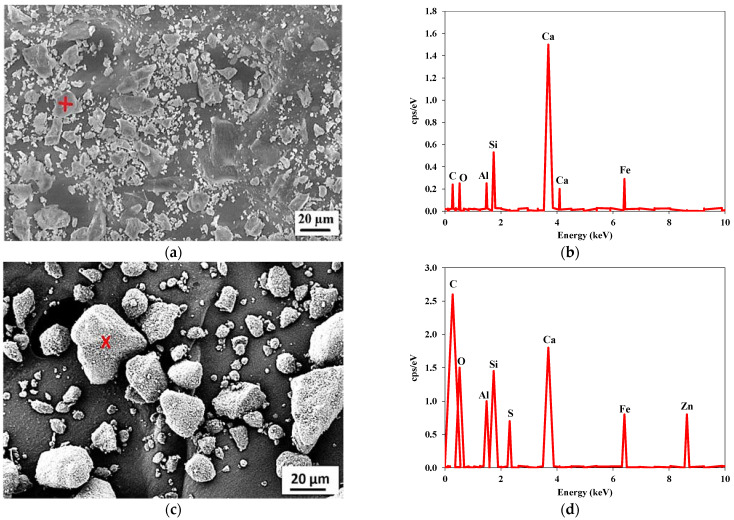
SEM and EDX analysis of used cement and rubber. (**a**) Cement; (**b**) EDX of cement (**+**); (**c**) Rubber waste; (**d**) EDX of rubber waste (**x**).

**Figure 4 materials-15-03584-f004:**
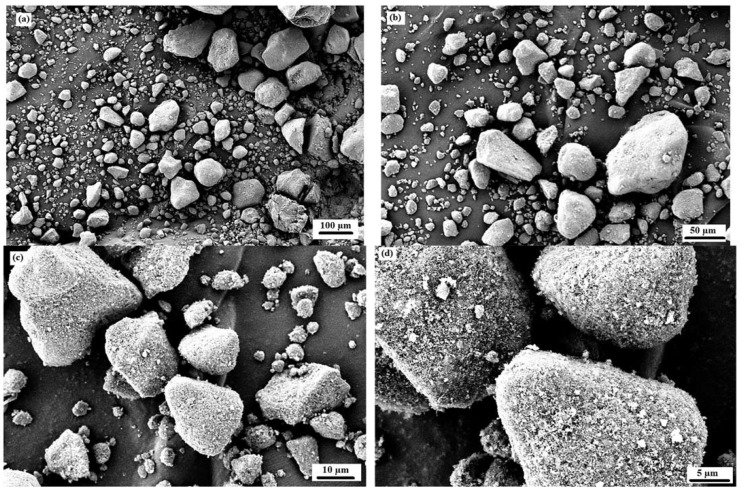
Micro-structure of rubber waste at various magnifications. (**a**) 100 µm; (**b**) 50 µm; (**c**) 10 µm; (**d**) 5 µm.

**Figure 5 materials-15-03584-f005:**
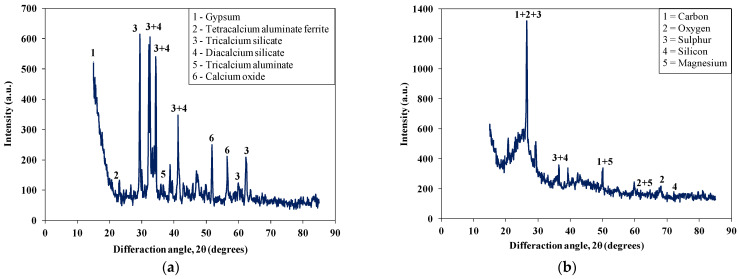
XRD patterns of raw materials. (**a**) Cement; (**b**) Rubber waste.

**Figure 6 materials-15-03584-f006:**
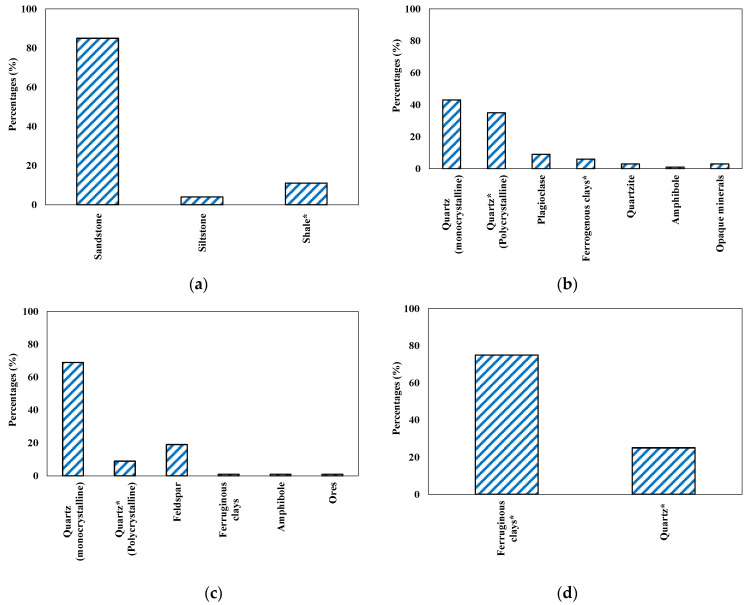
Petrographic analysis of used aggregate. (**a**) Main constituents of aggregate; (**b**) Sandstone; (**c**) Siltstone; (**d**) Shale (* indicate the reactive minerals).

**Figure 7 materials-15-03584-f007:**
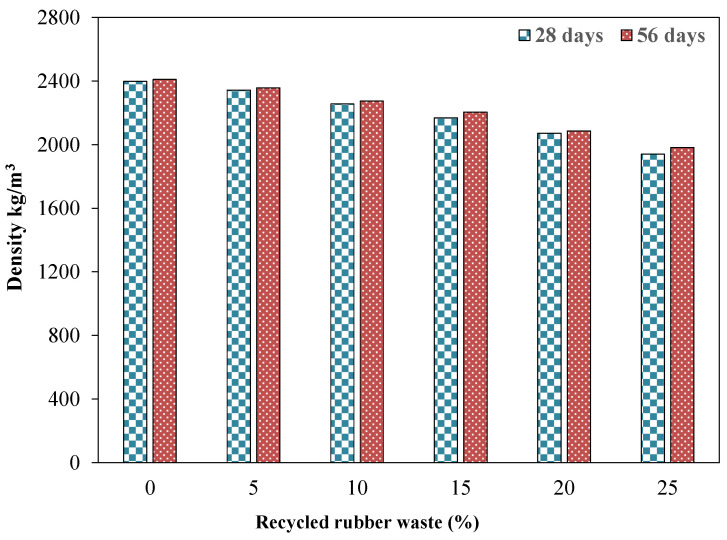
Density of specimens incorporating various dosages of RW at various ages.

**Figure 8 materials-15-03584-f008:**
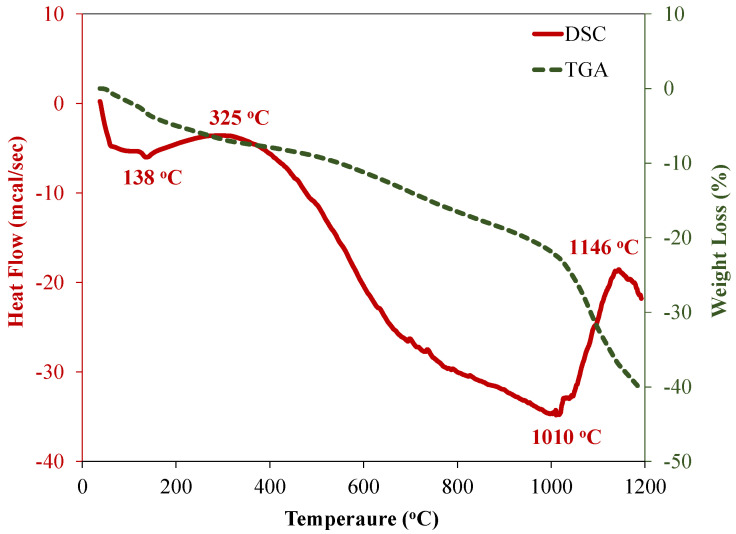
Thermal analysis of recycled rubber waste.

**Figure 9 materials-15-03584-f009:**
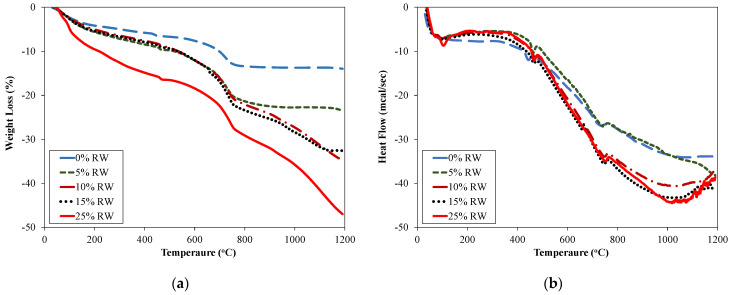
Thermal analysis results for specimens incorporating RW.(**a**) Weight loss; (**b**) Heat flow.

**Figure 10 materials-15-03584-f010:**
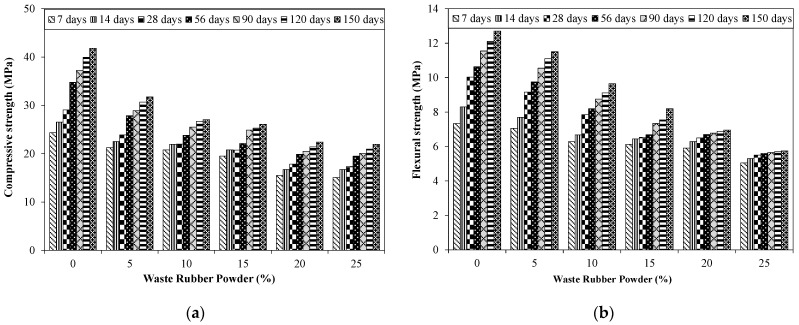
Effect of RW on compressive and flexural strength results. (**a**) Compressive strength; (**b**) Flexural strength.

**Figure 11 materials-15-03584-f011:**
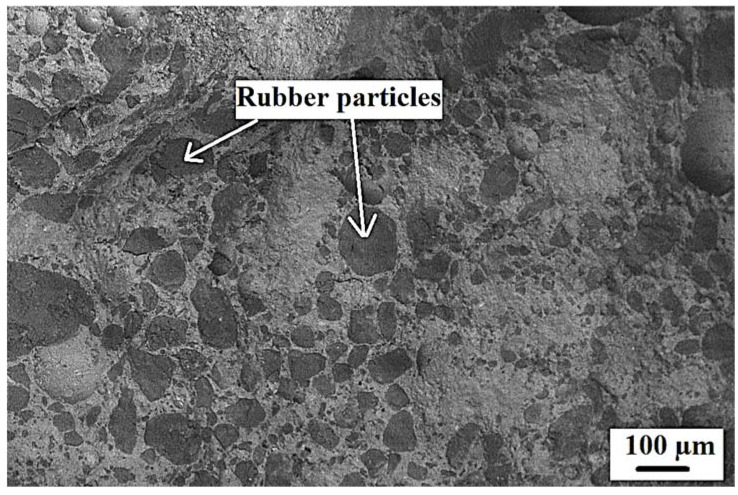
SEM image of concrete specimen incorporating 25% of RW.

**Figure 12 materials-15-03584-f012:**
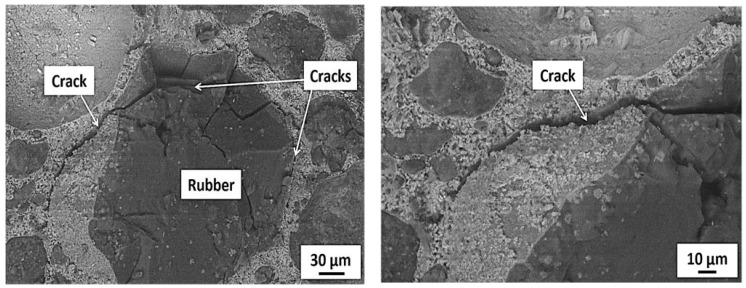
SEM image of control specimen incorporating RW.

**Figure 13 materials-15-03584-f013:**
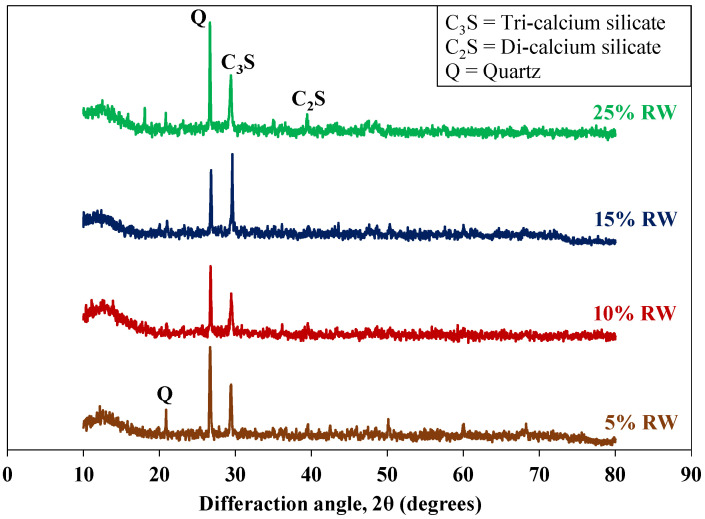
XRD pattern for specimens incorporating RW.

**Figure 14 materials-15-03584-f014:**
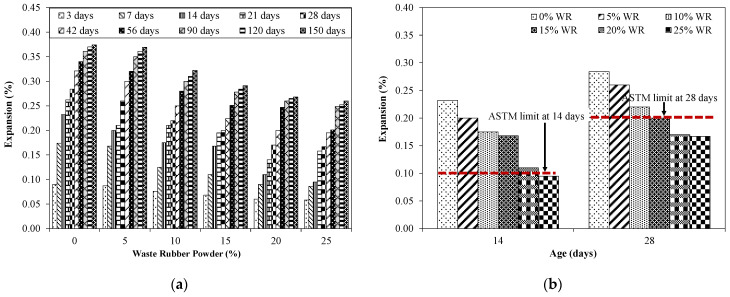
Effect of RW on expansion results and ASTM C1260 limits. (**a**) Expansion results; (**b**) ASTM limits for expansion.

**Figure 15 materials-15-03584-f015:**
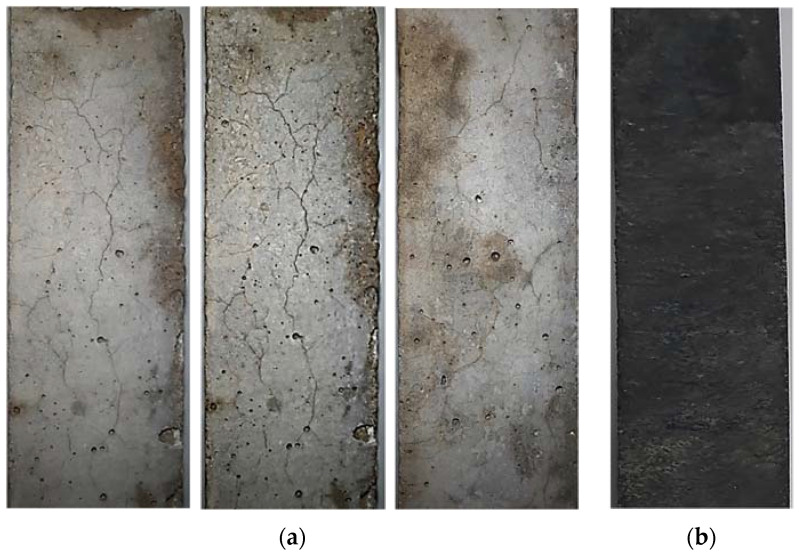
Surface cracks after exposure in ASR solution after 120 days. (**a**) Control specimens without RW; (**b**) Specimen with RW.

**Figure 16 materials-15-03584-f016:**
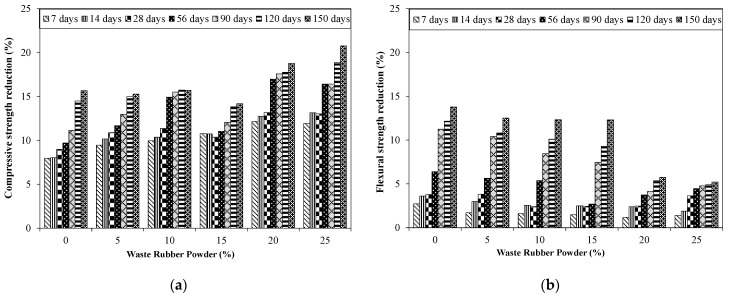
Compressive and flexural strength results of specimens incorporating RW exposed to ASR conditions. (**a**) Compressive strength; (**b**) Flexural strength.

**Figure 17 materials-15-03584-f017:**
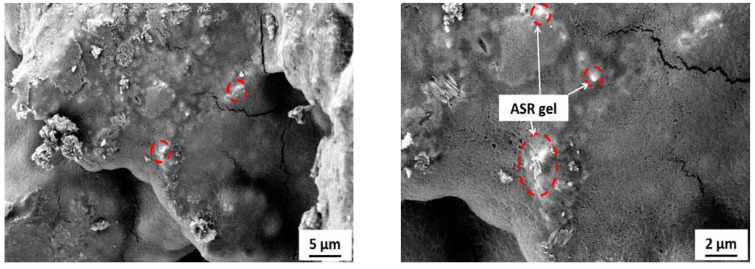
Formation of ASR gel in specimens exposed to ASR environment. (ASR gel location is shown by red dashed circles).

**Figure 18 materials-15-03584-f018:**
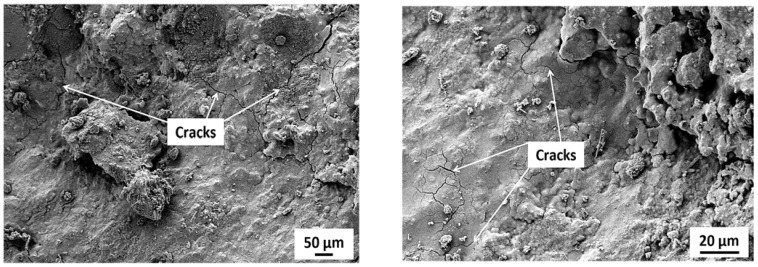
Micro-cracks in control specimens without RW exposed to ASR conditions.

**Figure 19 materials-15-03584-f019:**
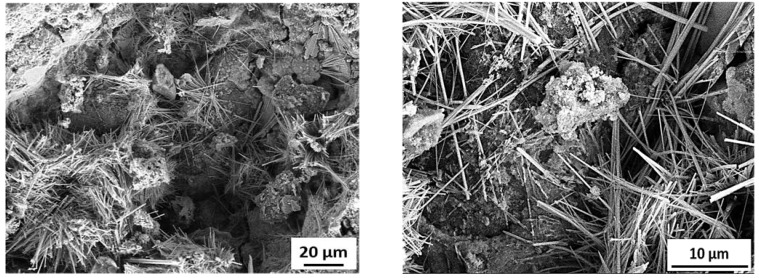
SEM image of specimen with 25% of RW exposed to ASR conditions.

**Table 2 materials-15-03584-t002:** Test matrix incorporating various proportions of recycled rubber waste.

Mixtures	Cement (%)	RW (%)
Control	100	0
RW5	95	5
RW10	90	10
RW15	85	15
RW20	80	20
RW25	75	25

RW = Rubber waste.

**Table 3 materials-15-03584-t003:** Chemical properties of rubber waste powder.

Elements	Percentage (%)		
	Present Study	Bisht and Ramana [53]	Angelin et al. [54]
Carbon (C)	74.83	87.50	91.50
Silicon (Si)	9.90	0.20	-
Aluminium (Al)	2.15	0.08	-
Zinc (Zn)	0.82	1.77	3.50
Magnesium (Mg)	0.26	0.14	-
Sulphur (S)	0.87	1.07	1.20
Oxygen (O)	11.17	9.24	3.30

**Table 4 materials-15-03584-t004:** Chemical properties of used cement.

Components	Result (%)
CaO	61.92
MgO	1.96
SiO_2_	20.67
SO_3_	2.59
Al_2_O_3_	4.98
Fe_2_O_3_	3.25
K_2_O	0.73
Na_2_O	0.10
LOI	2.61

**Table 5 materials-15-03584-t005:** Physical properties of cement and recycled rubber waste.

Properties/Tests	Cement	Recycled Rubber Waste
Specific gravity	3.15	1.24
Unit weight (kg/m^3^)	1427	285
Blaine fineness (cm^2^/g)	3075	3944
Fineness (Passing 200 sieve) (%)	>95	>93
Autoclave expansion (%)	0.13	-
Standard consistency (%)	24.6	-
Setting time (Initial, minutes)	120	195 ^¶^
Setting time (Final, minutes)	230	310 ^¶^

^¶^ For mixture incorporating 15% of RW.

**Table 6 materials-15-03584-t006:** Physical and chemical properties of used aggregates.

Tests/Components		Results
	Specific gravity	2.63
Physical	Bulk density (kg/m^3^)	1429
	Voids content (%)	40.23
	Impact value (%)	20.81
	Abrasion test (%)	22.44
	Water absorption (%)	2.11
Chemical	Silica (%)	75.45
	Calcium oxide (%)	3.80
	Magnesium oxide (%)	2.10
	Alumina (%)	3.92
	Ferric oxide (%)	4.84
	Loss on ignition (%)	7.24

**Table 7 materials-15-03584-t007:** Flow results of tested mixtures.

Mixtures	Flow (mm)
RW0	114
RW5	110
RW10	108
RW15	105
RW20	102
RW25	101

## Data Availability

Not applicable.

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
