# Peer review of "Recycled Untreated Rubber Waste for Controlling the Alkali–Silica Reaction in Concrete"

_materials, 2022, doi:10.3390/ma15103584_

Round 1
Reviewer 1 Report
The manuscript is defiantly shown a great effort of experiments, and it is worth to be published. However, I would like to address the following questions or comments to be taken into consideration when revising the manuscript:
1- Authors should improve the quality of all figures.
2- I do not see any characterization tests for mixtures (for example: shaking table tests?) It is essential to watch how the mixture behaves.
3- The details of specimen preparation are essential. For example, figures of the test specimens preparation process, please add them.
4- Are the dimensions of the molds associated with standards?
5- The conclusion must be reinforced.
6- It is better to have this paper extensively edited to improve the language.
7- Authors must be correct references according to the journal guidelines.
8- The introduction part is not fully cited, and a lot of cement research has been carried out, for example, Jiajian Li, Shuai Cao, Erol Yilmaz, and Yunpeng Liu. Compressive fatigue behavior and failure evolution of additive fiber-reinforced cemented tailings composites, Int. J. Miner. Metall. Mater. 2021.
In summary, the reviewer believes that this manuscript is recommended for publication through the abovementioned revisions.
Author Response
Please find the attached file for point by point response to the reviewer's comments.

Reviewer 2 Report
Dear Authors,
The manuscript is very interesting, but it needs a significant revision of the written English in terms of grammar and the lack of articles before nouns (e.g. Line 74 - "Similarly a study conducted by... showed a 50%..."). This occurs from the beginning till the end of the manuscript. It is imperative than an English-speaking person must revise the text of the manuscript.
In the attached manuscript follow some more comments and revisions in order to improve your manuscript.
Best regards

Author Response

(The authors gave the same response as above.)

Round 2
Reviewer 1 Report
Now, this revised manuscript can be accepted in its present form.